# Design of Optimized Coded LFM Waveform for Spectrum Shared Radar System

**DOI:** 10.3390/s21175796

**Published:** 2021-08-28

**Authors:** Dong-Hoon Kim, Hyung-Jung Kim, Jae-Han Lim

**Affiliations:** 1Department of Information and Telecommunication Engineering, Incheon Nat’l University, Incheon 22012, Korea; dhkim85@inu.ac.kr; 2Electronics and Telecommunications Research Institute, Daejeon 34129, Korea; acekim@etri.re.kr; 3Department of Software, Kwangwoon University, Seoul 01897, Korea

**Keywords:** Spectrum Shared Radar Systems (SSRS), orthogonal waveforms, code diversity, optimized coded LFM

## Abstract

To meet the increasing demands for remote sensing, a number of radar systems using Linear Frequency Modulation (LFM) waveforms have been deployed, causing the problem of depleting frequency resources. To address this problem, several researchers have proposed the Spectrum Shared Radar System (SSRS) in which multiple radars share the same frequency band to transmit and receive their own signals. To mitigate the interferences caused by the signal transmission by other radars, SSRS employs orthogonal waveforms that inherit the orthogonality of the waveforms from orthogonal codes. However, the inherited orthogonality of the codes is significantly reduced when incorporating LFM waveforms with the codes. To solve this problem, in this paper, we propose a novel but simple scheme for generating a set of optimized coded LFM waveforms via new optimization framework. In the optimization framework, we minimize the weighted sum of autocorrelation sidelobe peaks (ASP) and cross-correlation peaks (CP) of the coded LFM waveforms to maximize the orthogonality of the waveforms. Through computer simulations, we show that the waveforms generated by the proposed scheme outperform the waveforms created by previous proposals in terms of ASP and CP.

## 1. Introduction

A Linear Frequency Modulation (LFM) waveform is the most popular type of pulse compression and has been adopted for many radar systems due to its ease of hardware implementation and good range resolution. However, due to the increasing demands for mobile communications and remote sensing, a frequency resource that could be allocated to the radar systems and that employs an LFM waveform is lacking. To remedy this issue, many researchers have proposed the use of orthogonal waveforms in modern radar systems. The representative system adopting orthogonal waveforms is the Spectrum Shared Radar System (SSRS), where multiple radars in close proximity share the same frequency band as depicted in Figure 1 [1,2,3]. To estimate a target’s characteristics accurately in the SSRS, each radar must operate independently using a waveform that does not interfere with other radar signals. For this purpose, it is imperative to design orthogonal waveforms that minimize the interferences.

A simple solution to minimize interference in SSRS is a time division approach. In this approach, only one radar transmits a waveform and receives an echo signal during one time period; after the period, one of the other radars operates in a given order. The limitation of this approach lies in the low utilization, which means that the other radars must wait until the operation of one radar is completed. The low utilization inevitably hinders the real-time detection and estimation of a target’s characteristics, such as its position and velocity.

To address the limitations of the time division approach, several researchers have employed orthogonal codes to generate orthogonal radar waveforms. H. Deng proposed an optimization framework for generating a polyphase code set through the hybrid optimization process [4]. Specifically, we can derive a polyphase code set that minimizes the autocorrelation sidelobe peak (ASP) and cross-correlation peak (CP) via the optimization framework. In [5], U. Majumder et al. proposed the incorporation of an orthogonal code (e.g., Walsh–Hadamard code) into the Linear Frequency Modulation (LFM) waveforms via Spread Spectrum Coded LFM (SSCL). Specifically, an orthogonal waveform is created by multiplying an LFM waveform with each element of an orthogonal code. Even if the method for creating the SSCL waveform is simple, this method has a limitation in that multiplication with the rapidly fluctuated LFM waveform could significantly reduce the orthogonality of the code.

To address the problem mentioned above, we propose a novel but simple scheme for creating a set of orthogonal LFM waveforms that maximizes the orthogonality of the coded LFM waveforms. Here, the coded LFM waveform refers to an LFM signal encoded with a code. To maximize the orthogonality, our scheme employs a new optimization framework whereby we optimize coded LFM waveforms rather than orthogonal codes. In previous approaches, an orthogonal code set was firstly derived through an optimization framework (or the well-known orthogonal code was used) and then each code was multiplied with an LFM waveform. In contrast, we derive a set of orthogonal coded LFM waveforms through the proposed optimization framework. The beauty of the proposed scheme lies in the improvement of the orthogonality while benefiting from all the advantages of the LFM and an orthogonal code set. Our simulation results shows that our scheme exhibits significant performance improvements over the scheme proposed in [5]. This improvement comes from (1) the new optimization framework for generating optimized coded LFM waveforms and (2) a polyphase code with a higher degree of freedom compared to Walsh–Hadamard code.

We analyze the impact of several parameters (e.g., the number of distinct phases, the length of the code, and the number of radars) on the orthogonality of radar waveforms using computer simulations. Moreover, we observe that the proposed optimized coded LFM outperforms the waveform of [5] by up to 11.652 dB and 17.560 dB in terms of ASP and CP, respectively. The contributions in this paper are as follows:We propose a scheme for designing a set of optimized coded LFM waveforms to improve the orthogonality of previous proposals;The set of optimized coded LFM waveforms provides optimized orthogonality while inheriting various advantages of LFM;We propose an optimization framework for deriving the proposed optimal coded LFM waveforms;We conduct an in-depth study of the proposed optimal coded LFM waveforms through computer simulations and show that the proposed coded LFM waveforms outperform the orthogonal waveforms in previous proposals.

The remainder of this document is structured as follows. Section 2 briefly introduces the related works on orthogonal waveform design and highlights their limitations. In Section 3, we describe the details of the proposed scheme. Section 4 presents the simulation results and performance evaluations in terms of orthogonality in relation to ASP and CP. Numerical results and a discussion are provided in Section 5. Finally, Section 6 summarizes the results and draws conclusions.

## 2. Related Works

To maximize the system utilization in the SSRS, several researchers have proposed a mechanism for deriving an orthogonal waveform set by leveraging phase code diversity and chirp modulation diversity. Deng exploited phase code diversity to design an orthogonal polyphase code set that has good ASP and CP using a hybrid optimization process that leverages simulated annealing and the traditional iterative code selection method [4]. Afterwards, the authors in [6,7] adopted a genetic algorithm and the Niche genetic algorithm in the optimization process to improve the orthogonality of the code sets. In [8], the authors revealed a problem that the method could be trapped in the local minimum shortly after initialization, which resulted in poor performance when designing a polyphase waveform. To solve this problem, the authors proposed a relaxation and cyclic optimization algorithm.

Several researchers have proposed the use of chirp diversity to generate orthogonal waveforms [9,10,11,12,13,14]. Welstead studied the relationship between the chirp rate and the level of interferences between two LFM signals with the same pulse width and showed that, as the difference of the chirp rates of two LFM waveforms increases, they become less correlated [9]. Focusing on the characteristics of the chirp rate diversity of a single LFM signal, in [9,10], a method for suppressing cross-term interference using a modified LFM signal composed of multiple sub-chirp signals with different chirp rates was implemented. However, when two signals with the same pulse width have different chirp rates, the bandwidth of one signal is wider than that of the other signal. Like [10], in [11,12,13,14], the authors exploited chirp diversity to generate orthogonal waveforms using a signal composed of several sub-chirps with different chirp rates. Unlike previous studies [9,10], the authors in [11,12,13,14] exploited chirp diversity, but bounded the bandwidth so that all signals had the same bandwidth. In [11], the author developed a waveform consisting of up-chirps and down-chirps within the bounded bandwidth and repeated the waveform within the pulse width. The method for developing a waveform in [11] showed very good cross-correlation performance between a pair of waveforms with different chirp rates. However, the performance in terms of the autocorrelation sidelobe peaks of the waveform was unsatisfactory. The authors in [12,13] constructed a new class of a polyphase code set called piecewise LFM by concatenating the P3/P4 polyphase codes; they examined the autocorrelation side lobe peak and cross-correlation peak performance as well as the Doppler property. This piecewise LFM waveform is composed of several sub-chirps with different chirp rates and sub-chirp durations. Each sub-chirp is configured to occupy an independent bandwidth that does not overlap with other sub-chirps within the specified piecewise LFM bandwidth. In this manner, several orthogonal waveforms can be generated by controlling different sub-chirp durations and chirp rates. Compared to [4], this waveform showed good Doppler properties; however, the performance in terms of autocorrelation and cross-correlation was not significantly improved.

Multiple diversities have been considered for the design of a set of orthogonal waveforms. U. Majumder proposed a scheme for designing SSCL (Spread Spectrum Coded LFM) waveforms that include code diversity and chirp rate diversity [5]. These waveforms are generated by generating LFM signals with different chirp rates (chirp rate diversity) and encoding the LFM signals with an orthogonal code set (code diversity). However, the previous approach for generating a SSCL waveform [5] might reduce the orthogonality of the orthogonal code because each element of the code could be distorted when multiplied with a LFM waveform. To address this issue, we propose a scheme for deriving a set of optimized coded LFM waveforms that maximizes orthogonality. We elaborate on the proposed scheme in the following section.

Instead of designing orthogonal waveforms, several studies have focused on improving orthogonal waveforms by using a pulse train or modifying a matched filter. In [15], the authors mentioned the range–Doppler ambiguity problem of several optimal waveforms and proposed a method to generate optimal frequency-modulated pulse trains to alleviate this problem. In [16], an orthogonal waveform called DFCCW was implemented. The sub-pulses of the waveform proposed in [16] had different carriers and chirp rates, indicating that they had more degrees of freedom to control the waveform properties. This method has been proven to be excellent in reducing cross-ambiguity peaks because of the pulse train property. In [17], the authors showed that high recurrent Doppler sidelobes caused serious interference between channels. To suppress these recurrent side-lobes, they proposed a modified range–Doppler processing method at the receiver.

## 3. Proposed Scheme

To derive a set of optimized coded LFM waveforms, our scheme employs a new optimization framework in which we find L coded LFM waveforms minimizing an objective function, which is the weighted sum of ASP and CP. Hence, to understand the process of deriving a set of optimized coded LFM waveforms, we first explain the LFM signal and the set of polyphase codes used to encode the LFM signal. Then, we explain how to find a set of optimized coded LFM waveforms through the proposed optimization process. Before explaining the details, we summarize the nomenclature of this paper in Table 1.

### 3.1. LFM Waveform

An LFM waveform is a signal in which a frequency increases (up-chirp) or decreases (down-chirp) linearly with time, and the LFM waveform is obtained by
(1)$$x\left(t\right)=\exp\left[j2\pi \left(f_{0}t+\frac{1}{2}\alpha t^{2}\right)\right]\cdot 1_{\left[0,T_{p}\right]}\left(t\right)$$
where $T_{p}$, $f_{0}$, and α are the pulse width, starting frequency, and chirp rate, respectively. $1_{\left[0,T_{p}\right]}$ is an indicator function and is expressed as follows.
(2)$$1_{\left[0,T\right]}\left(t\right)=\left\{\begin{array}{c} 1,\;0\leq t\leq T\\ 0,\;otherwise. \end{array}\right.$$

When the starting frequency $\left(f_{0}\right)$ is zero, Equation (Equation 2) can be simplified as follows.
(3)$$x\left(t\right)=\exp\left(j\pi \alpha t^{2}\right)\cdot 1_{\left[0,T_{p}\right]}\left(t\right)$$

An LFM waveform set, X, consists of *L* LFM waveforms and is defined as follows (each LFM waveform in X might have different chirp rates. If this is the case, a modified LFM signal such as a concatenated LFM or piecewise LFM could be used to make the bandwidth of each LFM signal the same. In this case, chirp rate perturbation can be activated in addition to phase perturbation in the optimization process):(4)$$X={\left[x_{0}\left(t\right),x_{1}\left(t\right),\cdots ,x_{L-1}\left(t\right)\right]}^{T}$$
where $V^{T}$ is the transposition of a vector V.

### 3.2. Polyphase Code Set

To generate a coded LFM waveform, we employ a polyphase code sequence that allows a higher degree of freedom in our optimization process than a well-known binary code set (e.g., Barker code, Walsh–Hadamard code, Gold sequence), leading to better orthogonality (instead of a polyphase code set, other code sets can be used without any modifications in our design). A polyphase code set is composed of *L* codes; each code includes *N* sub-pulses elements. The polyphase code set S is defined as follows:(5)$$S=[s_{0},s_{1},\cdots ,s_{L-1}]$$
where $s_{l}$ is a code *l* and $s_{l}\left(n\right)$ is the *n*th element of $s_{l}$, which can be expressed by
(6)$$s_{l}=\left[s_{l}\left(0\right),s_{l}\left(1\right),\cdots ,s_{l}\left(N-1\right)\right]$$
and
(7)$$\{s_{l}\left(n\right)=\exp\left[j\phi_{l}\left(n\right)\right],\;n=0,1,\cdots ,N-1\},l=0,1,\cdots ,L-1$$
where $\phi_{l}\left(n\right)\left(0\leq \phi_{l}\left(n\right)\leq 2\pi \right)$ is the phase of $s_{l}\left(n\right)$. It is noted that the *n*th element of code *l* is used to encode the *n*th sub-pulse of coded waveform *l*. $\phi_{l}\left(n\right)$ can be selected from the following admissible values:(8)$$\phi_{l}\left(n\right)\in \left\{0,\frac{2\pi }{M},2\cdot \frac{2\pi }{M},\cdots ,\left(M-1\right)\cdot \frac{2\pi }{M}\right\}$$
where *M* is the number of distinct phases in a code.

### 3.3. Coded LFM Waveform

The *l*th coded LFM waveform, $y_{l}\left(t\right)$, can be generated by multiplying the *l*th LFM waveform and *l*th code sequence and can be defined as
(9)$$\begin{array}{cc} y_{l}\left(t\right) & =x_{l}\left(t\right)\cdot \sum_{n=0}^{N-1}1_{\left[0,T_{c}\right]}\left(t-nT_{c}\right)\cdot s_{l}\left(n\right)\\ \; & =\sum_{n=0}^{N-1}e^{j\pi \alpha_{l}t^{2}}\cdot 1_{\left[0,T_{c}\right]}\left(t-nT_{c}\right)\cdot e^{j\phi_{l}\left(n\right)} \end{array}$$
where $T_{c}$ is the chip time of a code sequence. $T_{c}$ is generally shorter than the pulse time, $T_{p}$, and is defined as $T_{c}=T_{p}/N$. That is, the original pulse is split into N sub-pulses, and each code element of the generated code sequence is encoded in each sub-pulse. We briefly define Equation (Equation 9) as
(10)$$y_{l}\left(t\right)=x_{l}\left(t\right)⊙s_{l}.$$

A set of coded LFM waveforms, Y, consists of *L* coded LFM waveforms as follows:(11)$$\begin{array}{cc} Y & =[y_{0}\left(t\right),y_{1}\left(t\right),\cdots ,y_{L-1}\left(t\right)]\\ \; & =X\circ S \end{array}$$
where X∘S is an element operation of ⊙ between X and S.

### 3.4. Optimization Framework

If S is first optimized and then generates a coded LFM waveform based on the optimized S, the orthogonality of S would be degraded. As a solution to this problem, we propose a new optimization framework in which we find a set of optimized coded LFM waveforms, $Y_{\mathrm{opt}}$, that minimizes an objective function; that is, the weighted sum of ASP and CP. As we do not vary an LFM waveform set in this optimization framework, finding $Y_{\mathrm{opt}}$ is comparable to finding $S_{\mathrm{opt}}$. Hence, we use S as a configurable parameter of our optimization framework. Through this approach, we can maximize the orthogonality of coded LFM waveforms, which are actually transmitted by radar.

The optimization problem can be formulated as follows:(12)$$S_{\mathrm{opt}}=\underset{S}{\arg\min}Obj\left(Y\right)$$
where Obj(Y) is an objective function when a set of coded LFM waveforms is Y. Obj(Y) can be expressed by
(13)Obj(Y)=(1−λ)ASP(Y)+λCP(Y)
where λ is a weighting factor (it is noted that the increase in λ means that we design a set of coded LFM waveforms to mitigate CP more aggressively than ASP), and ASP(Y) and CP(Y) are the autocorrelation sidelobe peak and cross-correlation peak, respectively. ASP(Y) and CP(Y) are described by
(14)$$ASP\left(Y\right)=\max_{i}\max_{|k|>ML_{i}}\left|R_{i}\left(k\right)\right|$$
(15)$$CP\left(Y\right)=\max_{i,j,i\neq j}\;\max_{k}\;\left|R_{ij}\left(k\right)\right|$$
where *k* and $ML_{i}$ are the time delay and normalized main lobe width of the *i*th coded LFM waveform, respectively. $R_{ij}\left(\cdot \right)$ is a cross-correlation function of the *i*th and *j*th coded LFM waveforms and R(·) is an autocorrelation function of the *i*th coded LFM waveform. $R_{ij}\left(\cdot \right)$ can be expressed by
(16)$$R_{ij}\left(k\right)=\sum_{n=-\infty }^{k}y_{i}\left(n\right)\cdot y_{j}^{*}\left(n-k\right)$$
where $y_{i}\left(n\right)$ is the *n*th element of a discrete sequence for $y_{i}\left(t\right)$ and $y^{*}$ is a complex conjugate of *y*. $R_{i}\left(k\right)$ can be easily obtained by replacing $y_{j}^{*}\left(n-k\right)$ with $y_{i}^{*}\left(n-k\right)$. In radar systems adopting coded waveforms and matched filtering, the autocorrelation sidelobe (i.e., $R_{i}\left(k\right),k>0$) might induce a spurious peak, causing false alarms in target detection. Thus, maximizing $R_{i}\left(0\right)$ is comparable to minimizing ASP(Y). As it is impossible to minimize both ASP and CP in the optimization framework, we minimize the weighted sum of ASP and CP in our optimization framework. Similar to [13], the main lobe width $ML_{i}$ can be obtained by
(17)$$ML_{i}=\left[\min\left(k>0\right)\;s.t.\;|R_{i}\left(k\right)|<\frac{1}{\sqrt{N}}\right]-1.$$

As depicted in Figure 2, the procedure for finding $S_{opt}$ consists of six steps (notably, even if it seems that finding $Y_{\mathrm{opt}}$ is computationally complex, the complexity of finding $Y_{\mathrm{opt}}$ in this framework is comparable to the complexity of finding $S_{\mathrm{opt}}$ [6]). Specifically, we adopt simulated annealing (SA), as presented in [4], as an algorithm for finding an optimal point, which corresponds to steps 3–6. The detailed procedure is explained below.

Generate an LFM waveform set, X;Generate a random polyphase code set S;Generate a set of coded LFM waveforms, Y;Calculate the objective function, Obj(Y). Then, Y is adopted as a candidate of the optimized coded LFM waveform. If Obj(Y) is smaller than $Obj(Y_{opt})$, $Y_{opt}$ and $S_{opt}$ are replaced with the current Y and S. Otherwise, we do not change $Y_{opt}$ and $S_{opt}$ (any global optimization algorithm (e.g., brute force, simulated annealing, the genetic algorithm, Bayesian optimization, etc.) can be used in this step to find the optimal state. In this paper, we employ simulated annealing (SA), which was also adopted by [4]. In simulated annealing, the transition, which corresponds to the update of the phase matrices S and $S_{opt}$ in our context, is determined by an acceptable probability function $P\left(Obj\left(Y_{prev}\right),Obj\left(Y\right),T\right)=\exp\left[\left\{Obj\left(Y\right)-Obj\left(Y_{prev}\right)\right\}/T\right]$ that depends on the scores of the two states and a global time-varying parameter *T* called temperature. If a random value *p*, where p∼U(0,1) is a uniform distribution between 0 and 1, is less then $P(Obj\left(Y_{prev}\right),Obj\left(Y\right),T)$, then the transition is accepted for the current state, Y, as a candidate of the optimized coded LFM waveform, $Y_{opt}$. This probability prevents the algorithm from becoming stuck at a local minimum that is worse than the global minimum. Please refer to [4] for detailed operation);In the phase perturbation process, a randomly selected element in S is replaced with a different admissible phase, which is defined in Equation (Equation 8);Repeat steps 3 to 5 until the stop condition is satisfied (we use the stop conditions adopted by [4]. If the phase perturbation is not accepted during three consecutive temperature reductions in the simulated annealing algorithm, then the optimization process is stopped).

## 4. Performance Evaluation

In this section, we present the evaluation of the proposed scheme for generating a set of optimized coded LFM waveforms using MATLAB R2020b [18]. More specifically, we obtained two performance measures to evaluate the orthogonality (i.e., ASP and CP) as a function of several independent variables: the code length (*N*), the number of distinct phases (*M*), the weighting factor (λ), and the number of radars (*L*). As the optimization process is a task to find a phase matrix S, we selected *L*, *M*, and *N* to constitute the phase matrix S as independent variables, and λ was used for the objective function. The chirp bandwidth and pulse width were fixed to 16 MHz and 1 μs, respectively (the netted radar system for detecting drones normally consists of up to 8 radars and uses the same pulse length of 1 μ to perform the same role [19] and 16 MHz, respectively). The sampling rate used in the MATLAB simulation was set to 256 MHz. The bandwidth of the coded LFM is determined by the sum of the bandwidth of the LFM waveform (*B*) and the bandwidth of the code (1/Tc). This sampling rate satisfies the Nyquist bandwidth for the bandwidth of the coded LFM waveform used in the experiment.

To obtain a set of optimized coded LFM waveforms in this evaluation, we adopted a hybrid optimization process similar to [4]. To demonstrate the superiority of the proposed scheme, we compared the orthogonality of the proposed optimized coded LFM waveforms with that of SSCL waveforms [5]. In the SSCL waveforms, we generated two types of coded waveforms: (1) SSCL waveforms encoded with the Walsh–Hadamard code (SSCL–Hadamard) and (2) SSCL waveforms encoded with the polyphase codes derived in [4] (SSCL–Polyphase). The setting of default parameters is shown in Table 2.

### 4.1. Impact of the Number of Radars

Figure 3 depicts ASP and CP according to the number of radars. We observe that the ASP and CP of the proposed optimized coded LFM waveforms were lower than those of the SSCL–Hadamard and SSCL–Polyphase waveforms. This was because the objective function of the proposed optimization framework was a function of coded LFM waveforms. In contrast, the objective function for obtained SSCL–Hadamard and SSCL–Polyphase waveforms was based on orthogonal codes, where orthogonality could be degraded during multiplication with LFM waveforms.

Figure 3 shows that ASP and CP increased with the number of radars in both the proposed coded LFM waveforms and two benchmarks (SSCL–Hadamard and SSCL–Polyphase). This was because it was unlikely that a set of waveforms would be selected that could produce good orthogonality for all pairs of waveforms as *L* increased. Moreover, in Figure 3, an additional gain was obtained by increasing the number of distinct phases in the proposed LFM waveforms.

### 4.2. Impact of Code Length

Figure 4 shows ASP and CP according to the code length. As expected, the optimized coded LFM waveform outperformed the SSCL–Hadamard and SSCL–Polyphase waveforms in terms of ASP and CP. Similar to Figure 3, Figure 4 shows that the additional gain could be obtained by selecting a large *M* in the proposed optimized coded LFM waveform. Specifically, CP decreased exponentially with the code length and ASP was likely to converge with the code length. From this observation, we conclude that the code length is a dominant factor affecting the orthogonality. This is because the degree of freedom (DoF) increases exponentially with the code length, which implies that we can select a set of coded waveforms with large orthogonality in the optimization process. From Figure 4, we learn that an appropriate tradeoff is required between the bandwidth of a radar system and the code length satisfying the orthogonal performance in the proposed scheme. Specifically, the bandwidth of the waveform increases linearly with the code length, but the gains in ASP and CP tend to gradually decrease with the code length, as depicted in Figure 4. This implies that a longer code length allows the orthogonal gain to be achievable with a greater number of phases.

### 4.3. Impact of the Number of Distinct Phases

Figure 3 and Figure 4 confirm that our scheme could generate a better set of orthogonal waveforms than the two benchmarks. We expected that we could further improve the orthogonal waveforms of our scheme by controlling the number of distinct phases. To confirm this, we evaluated the ASP and CP of the proposed optimized LFM waveforms according to the number of distinct phases. Figure 5 shows that both ASP and CP decreased with the number of distinct phases. In particular, the gain induced by adopting a large *M* increased with the code length *N*. This phenomenon resulted from the fact that the degree of freedom (DoF) in the optimization process increased exponentially with *N* (the number of coded LFM waveforms that could be selected in the optimization process was $M^{L\times N}$).

### 4.4. Impact of the Weighting Factor

Recall that the weighting factor λ is an important design parameter affecting the ASP and CP of optimized coded LFM waveforms. Specifically, a set of coded LFM waveforms with a large λ reduces CP aggressively while sacrificing ASP. Thus, we evaluated the ASP and CP of the optimized coded LFM waveforms of our scheme according to λ.

Figure 6 depicts ASP and CP according to a weighting factor λ. As expected, CP decreases with λ and ASP increases with λ. There are two interesting observations in Figure 6. First, ASP slightly increases with λ in the regime with a small λ(≤0.75) and abruptly increases with λ in the regime of a large λ(>0.75). However, CP slightly decreases with λ in both regimes. Specifically, the increase in λ in the regime of a large λ induces a small gain in CP but significantly degrades ASP. Second, we do not find any tendency between ASP and *M* when λ is 1. However, we observe that a large *M* is helpful for reducing CP. This is because ASP is not considered in the optimization process when λ is 1.

## 5. Discussion

In Table 3, we compare the performances of our scheme with those of previous proposals [6,7,13]. More specifically, the proposed scheme outperformed the previous proposals even when our scheme employed a shorter code length and a larger number of radars than the previous proposals. We emphasize that the orthogonal waveforms from our scheme showed better orthogonality than those from previous proposals even using a narrower bandwidth and supporting more radars. In other words, the frequency efficiency of our scheme is much better than previous proposals [6,7,13].

It should be noted that we must consider both the computational complexity and orthogonality when generating orthogonal waveforms using the proposed scheme. Specifically, large numbers of *M* and *N* induce a large DoF in the optimization process, but the computational complexity of the optimization process increases exponentially with *M* and *N*. Moreover, the coded LFM waveform with a large *M* is vulnerable to noise. Therefore, the careful selection of design parameters (e.g., *L*, *M*, *N*) is important for generating a set of optimized coded LFM waveforms.

To exploit the proposed waveform in practice, we need to analyze the performance of the range and Doppler measurements of the optimized coded LFM waveform compared to a single (uncoded) LFM waveform. For this purpose, using MATLAB, we obtain ambiguity functions of the proposed optimized coded LFM and a single (uncoded) LFM waveform, where the ambiguity function at a non-zero delay and Doppler frequency corresponds to returns from the range and Doppler that are different to those of the nominal target [20]. The LFM waveform is constructed with a pulse width of 1 μs and sweep bandwidth of 32 MHz. To generate an optimized coded LFM waveform set that has the same bandwidth as the LFM waveform, we encoded polyphase code sequences with a chip bandwidth of 16 MHz to an LFM waveform sweeping 16 MHz for 1 μs.

Figure 7 shows the range–Doppler result obtained through the zero-delay cut and the zero-Doppler cut ambiguity functions. As shown in Figure 7, the ambiguity functions of the non-zero delay of the proposed waveform are relatively higher than those of a single LFM waveform. This implies that accuracy of range measurements could be reduced when there are multiple targets in close proximity (we can improve the accuracy of range detection by employing a post-processing algorithm at the receivers (e.g., the CLEAN algorithm)). Moreover, the ambiguity function of the non-zero Doppler of the proposed waveform is comparable to that of a single LFM waveform, which means that the accuracy of Doppler measurement is rarely degraded.

## 6. Conclusions

In this paper, we propose a scheme for designing a set of optimized coded LFM waveforms. Contrary to previous approaches that derive a set of orthogonal codes that maximize the orthogonality of the codes and then use the derived codes to encode LFM waveforms, our scheme derives an optimal set of coded LFM waveforms, maximizing the orthogonality of the waveforms. For this purpose, our scheme employed a new optimization framework in which we found a set of coded LFM waveforms that minimize an objective function. Through in-depth simulation studies, we confirmed that the proposed scheme generated a set of waveforms with better orthogonality than previous proposals in terms of autocorrelation sidelobe peaks and cross-correlation peaks.

## Figures and Tables

**Figure 1 sensors-21-05796-f001:**
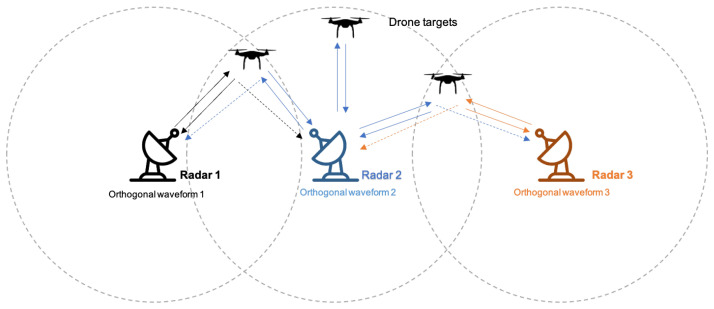
Spectrum Shared Radar System.

**Figure 2 sensors-21-05796-f002:**
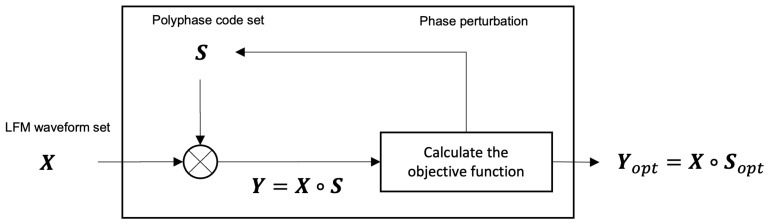
Structure of proposed coded LFM scheme.

**Figure 3 sensors-21-05796-f003:**
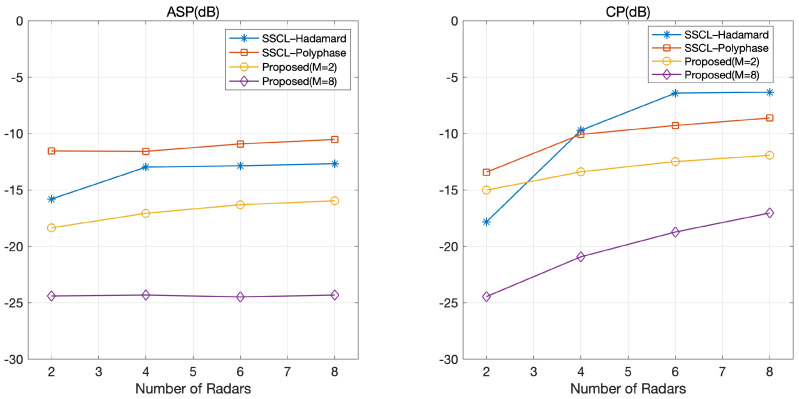
ASP and CP according to the number of radars.

**Figure 4 sensors-21-05796-f004:**
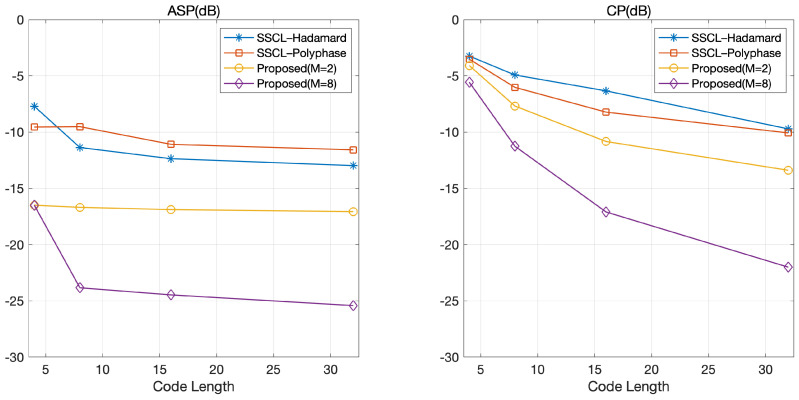
ASP and CP according to the code length.

**Figure 5 sensors-21-05796-f005:**
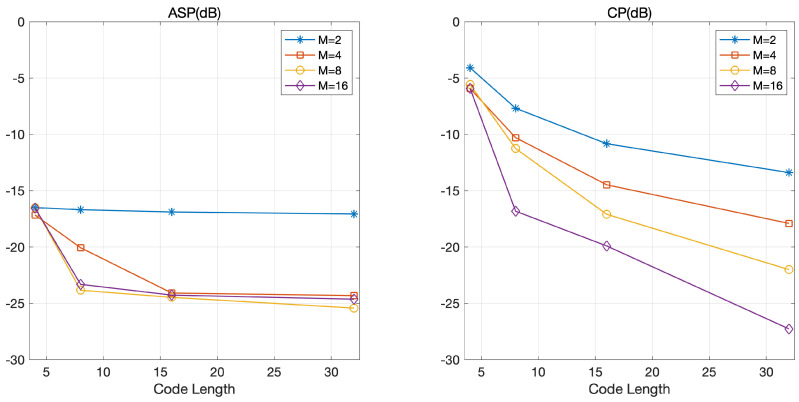
ASP and CP according to the code length in various settings for *M*: M=2, 4, 8, and 16.

**Figure 6 sensors-21-05796-f006:**
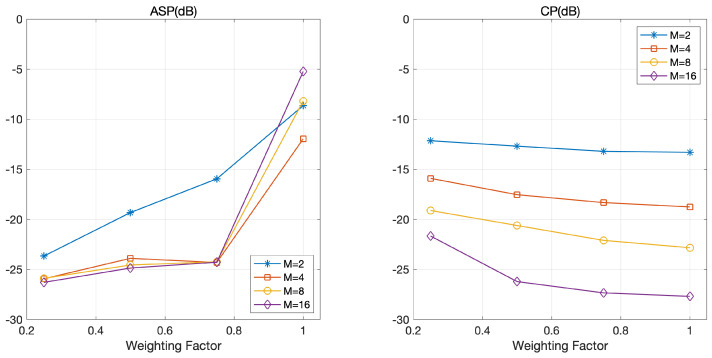
ASP and CP according to the weighting factor in various settings for *M*:M=2, 4, 8, and 16.

**Figure 7 sensors-21-05796-f007:**
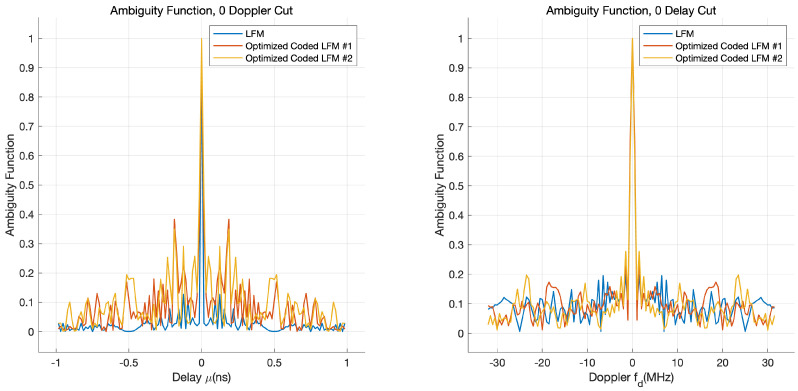
Range–Doppler relationship of a single LFM and optimized coded LFM waveforms.

**Table 1 sensors-21-05796-t001:** Nomenclature.

Symbol	Description
*L*	The number of radars or waveforms
*M*	The number of distinct phases
*N*	Code length
λ	Weighting factor
$T_{p}$	Pulse width of an LFM waveform
$T_{c}$	Chip time of a code sequence
X	A set of LFM waveforms
S	A set of polyphase code sequences
$S_{\mathrm{opt}}$	A set of optimized polyphase code sequences
Y	A set of coded LFM waveforms
ϕ	A phase element in S
ASP	Autocorrelation sidelobe peak
ML	Main lobe width
CP	Cross-correlation peak
$R_{i}\left(\cdot \right)$	Autocorrelation of the *i*th waveform
$R_{ij}\left(\cdot \right)$	Cross-correlation between *i*th and *j*th waveforms
Obj(Y)	The objective function that minimizes ASP and CP

**Table 2 sensors-21-05796-t002:** Parameter settings.

Parameter	Default Value
$T_{p}$, pulse width	1 μs
*B*, chirp bandwidth	16 MHz
*L*, the number of radars	4
*M*, the number of phases	2
*N*, code length	32
λ, weighting factor	0.5

**Table 3 sensors-21-05796-t003:** Performance comparison.

	Number of Radars	Code Length	ASP(dB)	CP(dB)
P. A. Qazi [13]	3	128	−17.66	−13.93
M. Ramarakula [6]	4	40	−17.8	−13.2
H. Zhang [7]	4	40	−15.2	−15.2
Proposed	8	32	−25.19	−21.88

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
