# Peer review of "Design of Optimized Coded LFM Waveform for Spectrum Shared Radar System"

_sensors, 2021, doi:10.3390/s21175796_

Round 1

Reviewer 1 Report

Spectrum Shared Radar System can use the same frequency band for transmitting and receiving their own signals. SSRS employs orthogonal waveforms that inherit the orthogonality of the waveforms from orthogonal codes.

The orthogonality of the codes can reduce when incorporating LFM waveforms, so, this paper proposes a scheme for generating a set of optimized coded LFM waveforms via new optimization framework.

The beauty of the proposed scheme lies in the improvement of the orthogonality while having all the advantages of the LFM and an orthogonal code set.

The authors marked as principal contributions: 1) the new optimization framework for generating optimized coded LFM waveforms; 2) a polyphase code having a higher degree of the freedom compared to Walsh-Hadamard code.

Comments:

  • In opinion of this reviewer, the authors should provide detailed review of each work referred in 2. Related Works section. This section is very short (of 0.5 page) to understand all drawbacks of the state-of-the art methods. Please provide revision of recently in 5-7 years published papers.
  • Please adjust the graphics presented in figs 3-6, presenting the experimental curves in solid lines and increasing letter sizes.
  • Please present detailed explications of the procedure for finding Sopt that consists of 6 steps. The authors should explain the algorithm and mathematically justify the phrases that are in algorithm (page 6): “calculate the objective function, Obj(Y), for any pair of Y. If finding better calculation result is obtained than the result of the previous iteration, the current polyphase code set, S, is adopted as the Sopt. In the phase perturbation process, randomly selected element in the S is replaced with a different admissible phase”. Such explication is not understandable for potential reader.
  • Please justify why did you select the methods from 4, 6, 7 references? Also, paper 4 is very old. In related works section, you mentioned methods 8-13 but did not provide comparison with these one, please justify such selection of comparison methods. Are there exist other recent methods in the literature?

Reviewer 2 Report

This paper proposes a scheme for designing a set of optimized coded LFM waveforms with better orthogonality than previous proposals in terms of autocorrelation sidelobe peaks and cross-correlation peaks. It may be applied to the spectrum shared radar system.

1. Theoretically speaking, the optimal orthogonal waveforms should have the maximum autocorrelation as well as the minimum cross-correlation. In the paper, the optimization framework is to minimize the weighted sum of autocorrelation sidelobe peaks and cross-correlation peaks of the coded LFM waveforms. What does it mean to optimize the autocorrelation sidelobe peaks with respect to autocorrelation?

2. The proposed coded LFM waveforms is generated by multiplying the LFM waveforms and the phase code sequence. The code sequence may disrupt the coherence of waveform in the CPI, which may result in a performance decrease in the Doppler coherent processing. The performance of ranging and Doppler measurement needs to be analyzed. The waveform design should take into account the waveform properties as well as the measure performance. The performance evaluation needs to be more comprehensive.

3. What is the relationship between Tp and Tc?

Reviewer 3 Report

Firstly, I admire the clarity of presentation of the subject of the paper. The Table 1 with nomenclature summary is useful.

I have a few minor comments only:

1) In Section 3.2 the "Gold sequence" should have capital G (named after Robert Gold)

2) Could you please elaborate on sample rate used in the Matlab simulation?

3) In Section 4: "A pulse width and the bandwidth of a LFM waveform are fixed to 1 us"
I believe the chirp bandwidth was fixed to 16 MHz and the pulse width to 1us.

4) I am missing a relation between LFM pulse width (T_p) and Chip time (T_c).

5) The rectangular pulse function (P(t)) is not properly defined, I believe its lenght should be specified (probably T_c).

Round 2

Reviewer 1 Report

No comments

Reviewer 2 Report

The previous concerns have been addressed in the revised manuscript, while the quality of the paper was improved.